**Beyond Cooperation: The Role of Origin Countries in Deportation Efforts, Evidence from Mexico (1942 to 1964)**

**Abstract**

For deportations to be carried out, host countries must secure inter-state cooperation with origin countries where they seek to deport noncitizens. However, origin countries may decide to cooperate or resist cooperation. This paper pushes the scholarship by exploring why some origin countries not only decide to cooperate but also become proactive actors in deportation efforts by encouraging and promoting the deportation of their citizens despite the economic and political costs. This paper unpacks this puzzle by analyzing why the Mexican government became a proactive actor in facilitating the deportation of its citizens from the 1940s to the 1960s. Drawing on state archival research in the U.S. and Mexico, I argue that the Mexican government became a proactive actor as a strategy to address its domestic challenges and to cultivate its diplomatic relations with the U.S. government and the benefits that came with such relation including the continuation of the Bracero Program, the largest guest worker program. This study makes two key contributions. Empirically, the paper offers an in-depth analysis into how Mexico, a country with the highest flows of deportation in the Western Hemisphere, managed inflows of deportations during one of the largest deportation operations in the U.S. Second through introducing the concept of proactiveness, it expands the field of migration diplomacy by showcasing how countries use deportation as a tool to advance their political priorities.

**Keywords**: Deportation, Mexico-U.S. migration, achieves, Bracero Program

## Introduction

On October 18, 1954, at 4 p.m., the SS *Emancipación*, an 1800-ton steel construction passenger

cargo vessel owned by the Mexican company Transportes Marítimos y Fluviales, S.C.L in

Mexico City[1] departed Port Isabel, Texas headed for Veracruz, Mexico. Inside the vessel were

800 Mexicans including men, women, and children. After sailing roughly sixty hours through the

Gulf of Mexico, the *Emancipación* arrived in Veracruz. Awaiting the Mexican deportees were

officials from Mexico's Immigration Services. After exiting the vessel, deportees were held in

---

[1] Letter from Joseph M. Swing, Commissioner of the Immigration and Naturalization Service (INS) to Joe M. Kilgore member of U.S House of Representatives, describing the SS *Emancipación* 1954, Research Group 85, Box 21936, National Archives, Washington, D.C.

warehouses and were processed by Mexican immigration officials. Repeaters who had previously been deported were taken into custody and sent to Mexico City or other across the interior of Mexico as a form of punishment. First time deportees were released in Veracruz.[2] This was one of the many voyages the SS *Emancipación* made to transport Mexican deportees from the U.S. Why did the Mexican government provide the U.S. government with the modes of transportation to deport its citizens and why did the Mexican government punish Mexicans by moving them into the interior of the country against their will despite the political and economic consequences?

The 1940s to the 1960s marked a period of rapid transformation in Mexico's domestic political economy and foreign policy. Domestically, the federal government shifted from supporting an agrarian economy to prioritizing large-scale industrialization (King 1970). However, such incentives were implemented in northern Mexico and disproportionately benefited commercially oriented agribusinesses (King 1970, 33). In terms of foreign relations, Mexico and the U.S. signed a series of bilateral agreements between 1942 and 1964 known as the Bracero Program. These agreements allowed both governments to establish common rules to administrate the hiring of Mexican workers through bilateral mechanisms (Delano 2011, 83).  For the Mexican government, the Bracero Program served as an opportunity to consolidate its diplomatic relations with the U.S. government following the expropriation of land from U.S. companies (see Dwyer 2008). However, Mexico's domestic and foreign policies also triggered an unprecedented number of unauthorized emigration to the U.S.

---

[2] Report written by Oran G. Pugh, U.S Border Patrol Inspector, about the boat lift from Port Isabel to Veracruz, October 27, 1954, Research Group 85, Box 21936, National Archives, Washington, D.C.

This paper examines why the Mexican government became a proactive actor in facilitating the deportation of its citizens from the U.S. between the 1940s and 1960s. I introduce the concept of proactiveness to capture instances in which origin countries extend their involvement beyond conventional inter-state cooperation in readmission processes. While standard cooperation typically involves verifying the nationality of deportees, issuing travel documents, and accepting forcibly returned nationals, proactiveness entails a more assertive role in determining the timing and modalities of deportation. For origin countries becoming a proactive actor includes actions such as encouraging the deportation of their citizens, providing logistical support to host countries—such as transportation, and implementing policies to deter individuals deported from re-emigrating. Importantly, proactiveness may encompass measure that are punitive in nature towards deported individuals from criminalizing to displacing deportees within their country of birth.

From the 1940s to the 1960s, the Mexican government not only participated in traditional readmission processes but also provided the U.S. government with mechanisms to facilitate the deportation of its citizens. Mexico requested that U.S. immigration authorities increase border vigilance, approved of private boat contracts to transport deportees, detained Mexicans who had previously been deported from the U.S., and relocated deportees into the interior of Mexico via trains, boat lifts and airlifts to deter re-emigration. The actions of the Mexican government during this period prompt us to ask why some origin countries are willing to go beyond conventional inter-state cooperation in readmission procedures and take on a proactive role— despite the political and economic costs, such as exposing deportees to violence and abuse by immigration officials, facing scrutiny from the media, to potentially disrupting the flow of remittances.

Drawing on state archival collections in the U.S. and Mexico, I argue that Mexico emerged as a proactive actor in U.S. deportation efforts from the 1940s through the 1960s—not only in response to U.S. pressure. Rather, the Mexican government strategically facilitated and promoted deportations to address its domestic priorities, including mitigating the loss of cheap agricultural labor in northern Mexico and strengthening its diplomatic ties with the U.S. The Mexican government became a proactive actor by transporting deportees under inhumane conditions, relocating them to interior regions far from their places of origin, and imprisoning repeat deportees. These measures were designed to deter re-emigration and to regulate the mobility of Mexican citizens in ways that protected the interests of domestic agribusiness in northern Mexico. In addition, Mexico's proactive stance strengthened its bilateral relationship with the U.S., especially during moments when U.S. immigration authorities framed rising apprehensions of Mexican nationals at its southern border as an 'invasion'. By assisting the U.S. in deporting its citizens, the Mexican government positioned itself as a key actor in implementing the U.S.'s deportation efforts and border policies. Finally, Mexico's proactiveness facilitated the continuation of the Bracero Program, which was crucial for offsetting unemployment in specific parts of Mexico and promoting development through the remittances sent by migrant workers.

This paper makes an empirical contribution to the study of Mexico–U.S. migration politics and advances the growing field of migration diplomacy (Thiollet 2011). Drawing on state archival collections in the U.S. and Mexico, it highlights Mexico's role in shaping the U.S.'s deportation efforts. The paper demonstrates that Mexico was not a passive actor in response to U.S. actions. By introducing the concept of *proactiveness*, the study seeks to expand the field of migration diplomacy—broadly defined as the ways in which countries use migrants as instruments and migration policies as formal and informal tools to pursue foreign and domestic policy objectives

(Thiollet 2011; Adamson and Tsourapas 2019). By foregrounding the concept of proactiveness, this study illuminates the diverse ways in which origin countries actively participate in deportation processes—to foster diplomatic ties with the countries that are seeking to deport noncitizens, but also to pursue their own domestic political objectives. The paper showcases how deportation practices are mutually constitutive, shaping both the countries that expel noncitizens and those that receive them, thereby complicating conventional understandings of power asymmetry.

This paper is structured as follows. The next section reviews the readmission agreement frameworks used by host and origin countries to carry out deportations and examines why origin countries either cooperate with or resist inter-state collaboration. The second section outlines the methodology, explaining why I consulted state archival collections in Mexico and the U.S., the nature of the sources examined, and what the archival data reveal about how and when deportation negotiations and processes occurred. The third section analyzes the political context in which Mexico became proactive in facilitating the deportation of its citizens and disaggregates the mechanisms through which it adopted a proactive role. The following section explores how Mexico's interests were shaped by domestic concerns, particularly at the regional and local levels. The final section examines how the Mexican government responded to U.S. deportation efforts and what insights the Mexican case from the 1940s to the 1960s offers for understanding the role of deportation in both domestic and global politics.

**To cooperate or not cooperate: The preferences of origin countries in managing deportations from host countries**

For deportations to be carried out successfully, host countries must secure bilateral or multilateral readmission agreements with the origin countries to which they seek to deport noncitizens. These agreements outline the protocols and responsibilities of both host and origin countries for implementing and processing deportations. Readmission agreements can be formal, informal, or a combination of both. Formal agreements typically undergo legislative and judicial review. Informal agreements, on the other hand, are often kept hidden from the public and consist of verbal or written exchanges between government officials without formal legal standing. This lack of legal status makes them difficult to subject to parliamentary or judicial scrutiny (Anderberg 2018). Scholars argue that, over the past decade, host and origin countries have increasingly leaned toward adopting informal readmission agreements because they offer 'low visibility' and are less likely to provoke political backlash or attract negative media attention (Cassarino 2007).

Studies demonstrate, however, that neither formal nor informal readmission agreements necessarily increase the rate of returns (Stutz and Trauner 2021), as origin countries vary in their willingness to cooperate (Cham and Adam 2023). Origin countries may either agree to cooperate or resist cooperation. Cooperation typically involves facilitating deportations by confirming the nationality of individuals subject to removal, issuing the necessary travel documents, and accepting deportees into the country. However, Zanker (2023) argues that there are multiple ways in which origin countries resist cooperation. These include reactive incompliance, which involves delaying the issuance of identification documents, and proactive incompliance, which refers to the outright refusal to cooperate, often by imposing moratoriums on deportation flights

(Zanker and Altrogge 2022). For example, since 2020, more than thirteen countries have resisted cooperation with the U.S. by refusing to accept their nationals in deportation proceedings. In 2019, The Gambian government implemented a moratorium on all deportation flights from both the European Union and the U.S. This reluctance to cooperate challenges the common assumption that origin countries with fewer economic resources will 'give in' to the demands of more powerful host countries. It also contests externalist perspectives, which claim that foreign governments and intergovernmental agencies are the primary actors shaping migration governance in origin countries (Gazzotti et al. 2023). Instead, such acts of resistance (Zanker 2023) have prompted scholars to explore how both domestic and foreign policies (Adam et al. 2020) influence inter-state cooperation in readmission procedures.

More scholars are gradually moving beyond power-based arguments that stress that countries in the global north have more leverage than countries in the global south in negotiating and implementing migration policies (Tsourapas and Adamson 2020; Gazzotti et al., 2023). Instead, scholars have argued that origin countries particularly those outside the global north are not 'merely passive actors' (Mouthaan 2019) to the demands of foreign actors but that domestic considerations and diplomatic relations with the deporting countries also shape their preferences in how they manage deportations. Adam and Cham (2023) showcase that origin countries may agree to cooperate not only because of external pressure, but also to appear legitimate and reliable to international actors. Origin countries therefore may decide to cooperate as a strategy not only to access materialistic resources but also intangible gains including status and recognition from external state and nonstate actors.

On the other hand, origin countries may contest cooperation based on domestic considerations. Zanker and Altragge (2022) argue that The Gambia placed a temporary moratorium on deportation flights from the EU in 2019 out of security and economic concerns. For the Gambia, deportees were perceived as a security threat, particularly in a context when the country was transitioning to a democratic regime. Another reason for resisting cooperation was the potential reduction in the inflow of remittances, which is an integral part of the country's GDP (Adam et. al., 2020). Moutheen (2019) also argues that origin countries may be less inclined to cooperate based on political implications, including political scrutiny and its impact on their relationship with diasporic communities. As scholars have demonstrated, diasporic and emigrant groups have gradually played an important role in the country's national political affairs and in the economies of their hometowns (Adida and Girod 2011). Scholars also highlight that origin countries may oppose cooperation with host countries as a form of neo-colonial resistance, to reject imperial legacies (McNeill 2023; Cham and Adam 2023), assert their political sovereignty and as a response to highlight the inequalities that prevail in international migration governance – favoring the interests of host countries in the global north (Ellerman 2008).

While the scholarship has made important contributions about why countries decide to resist or cooperate in readmission procedures, the Mexican case, in particular the state archival collections serve as a starting point for exploring when, how and why origin countries decide not only to cooperate or resist to the demands put forth by host countries, but under specific circumstances, encourage and promote the deportation of their citizens. Furthermore, the literature has assumed that deportation policies are only designed based on the interests of host countries (Ellerman 2003) and that host countries are the leading authors in designing

deportation policies. However, the Mexican case complicates such assumptions as it was the Mexican government that laid out possible ideas and insisted the U.S. government to control unauthorized border crossings and encouraged the removal of Mexicans through a combination of informal and formal agreements. Through the Mexican case, this paper seeks to motivate scholars to explore further how origin countries use deportation as a strategy to negotiate and address their domestic (national and subnational) concerns and foreign policy agendas. This study also seeks to unpack the conditions that motivate origin countries to become proactive actors in deportation proceedings to their political and economic advantage.

**Methodology**

To trace how and why the Mexican government became a proactive actor in the U.S.'s deportation efforts, I consulted state archival collections at the National Archives and Records Administration (NARA) in Washington, D.C. and at the *Archivo General de la Nación* (AGN), in Mexico City. I selected the NARA as it holds one of the largest collections from the Immigration Naturalization Services (INS) and the U.S. Border Patrol, the two agencies responsible for managing immigration enforcement and deportation operations across the country's interior and along the U.S.-Mexico border. On the Mexican side, I selected the AGN as it holds collections of policies and activities implemented under Mexican Presidential administrations.

It is important to acknowledge, as Varsanyi (2024, 85) notes, that state archival collections represent 'a selective record of the past that reflects the priorities of the state.' Following this

insight, I did not approach government-issued documents as sources of 'knowledge retrieval, but rather as instruments of knowledge production (Stoler 2002, 90)—specifically, knowledge about how and why Mexican federal authorities became proactively involved in supporting U.S. deportation operations despite potential political backlash. To address the research question, I selected sources that provided insights on the institutions and actors responsible for carrying out deportations, how deportation operations were designed and implemented, the demographics of deportees, and the interests shaping these policies. Therefore, the documents analyzed included diplomatic correspondence between U.S. and Mexican officials, reports from the INS and U.S. Border Patrol, formal and informal negotiation agreements, copies of contracts between government agencies and private entities, U.S. Congressional transcripts, internal memos between U.S. and Mexican immigration authorities, images of deportation operations, reports on deported individuals, and newspaper articles.

Through the documents I consulted, I was able to develop a sequence of events and develop 'thick descriptions' (Geertz 1973) of when and how deportations took place, as well as identify the actors and institutions in both the U.S. and Mexico involved in designing and implementing deportation policies. I was also able to unpack how negotiation processes unfolded and where key decisions were made (Hazareesingh and Nabulsi 2008, 151). These materials further allowed me to examine the conditions that motivated the Mexican federal government to become a proactive actor. Overall, the details gathered from these documents challenge assumptions about Mexico's passivity in U.S. deportation processes. The following sections explore the contexts in which the Mexican government adopted a proactive role in facilitating the deportation of its citizens.

**The Context: The Bracero Program (1942 to 1964) and managing cross-border mobility**

In the U.S., World War II created a demand for labor in agriculture and other industries. To address labor shortages, in 1942, the U.S. and Mexican governments signed a labor agreement allowing Mexican workers to enter the U.S. on short-term labor contracts. The agreements were intended to serve as a temporary wartime measure but were renewed until 1964. These agreements became known as the Bracero Program. Between 1942 to 1964, roughly 4.6 million Mexicans entered the U.S. with temporary visas to work predominately in the agricultural industry. Although the Mexican government was hesitant about entering into formal agreements with the U.S., it eventually viewed these agreements as a political and economic opportunity. Politically, the Bracero Program would permit the Mexican government to manage the flow of unauthorized labor emigration to the U.S., which the government had attempted but failed to accomplish since the early 20th century (Hernandez 2009; Fitzgerald 2006) to control the loss of a cheap labor force. As Delano (2011, 83) explains, the Bracero Program was 'exceptional' as it allowed both governments to establish 'common rules to administrate the hiring of Mexican workers through bilateral mechanisms.' Regarding economics, the Bracero Program would ease the domestic pressures of unemployment from rural parts of the country, and serve as an opportunity to modernize Mexico through the skills and remittances that braceros brought back (Craig 1971; Cohen 2001).  From the perspective of the Mexican government, the Bracero Program constituted a strategic mechanism for securing material gains—via the remittances braceros sent to their households—and intangible gains by positioning itself as an ally of the U.S. government through providing the nation with a cheap labor force during World War II.

During the early phase of the Bracero Program, Mexico had substantial leverage in the early stages of the negotiation process. However, from 1948 to 1951 the Bracero Program was decentralized. After WWII, Congress passed Public Law 40, officially terminating the Bracero Program. The Mexican government requested an extension because it was concerned about the potential implications of the mass return of braceros and growing unemployment rates in central western parts of Mexico. Although agreements between the U.S. and Mexican governments regarding the importation of laborers continued, the U.S. Congress did not endorse or provide oversight (Calavita 1992, 29) which meant that the U.S. government was no longer the employer of braceros. Instead, growers and their representatives contracted workers directly from recruitment centers in Mexico. Growers were responsible for hiring and transporting Mexican laborers. Eager to reduce transportation costs, growers pushed to relocate recruitment centers closer to the U.S.-Mexico border. However, the Mexican government resisted this move, arguing that proximity to the border would encourage unauthorized migration and further deplete the agricultural labor supply in northern Mexico.

Throughout the post-war period, the U.S. government and growers adopted a series of unilateral actions that went against the Mexican government's interests. Mexico's loss of power in the negotiation process was most evident during the 'El Paso' incident in October 1948, when the INS opened the border to Mexican workers for growers to hire on the spot. Roughly 7,000 to 8,000 men crossed the border under the surveillance of the INS from Ciudad Juarez, Chihuahua to El Paso, Texas. To address the flow of unauthorized migration, in 1949, in agreement with Mexican negotiators, the U.S. government adopted what Calavita calls a 'de facto legalization program' – a provision that became informally known as 'drying out the wetback' where the

U.S. government legalized undocumented Mexican workers who were already in the U.S. and were given preference for new contracts over newly imported braceros (Calavita 1992, 29). However, the provision was counterintuitive, as the flow of undocumented workers increased and failed to protect the rights of workers (Delano 2011, 93). From 1942 to 1952, roughly 2 million unauthorized Mexican migrants were apprehended by the INS.[3]

With the U.S. entering the Korean War (1950 to 1953), the Mexican government had a slight advantage in renegotiating agreements to continue the Bracero Program. The U.S.'s dependency on a cheap labor force and importation of raw resources from Mexico for military production gave Mexican President Miguel Alemán Valdés (1946 to 1952) some leverage in requesting the U.S. government to become the official employer of braceros. In 1951, the U.S Congress passed Public Law (PL) 78, which 'set the official parameters of the program' (Calavita 1992, 46) until the termination of the Bracero Program in 1964. Under PL 78, the U.S. federal government resumed direct responsibility for employing braceros. It also agreed to locate recruitment centers in Mexico's interior and to prohibit the hiring of undocumented workers. The implementation of PL 78 marked a turning point, as curbing unauthorized migration became a central priority for the U.S. government. These efforts were further reinforced by the political climate of the Cold War and McCarthyism, which intensified anti-immigrant sentiment and legitimized calls for greater control and securitization of the U.S.-Mexico border.

The press played a significant role in fueling xenophobic attitudes toward Mexican migrants (Goodman 2020; Garcia 1980). Reports by the INS and the media framed the rise in migrant

---

[3] Estimate was retrieved from Annual Reports of the Immigration and Naturalization Service, U.S. Department of Justice for Fiscal Year 1943 to 1952.

encounters at the U.S.-Mexico border as a 'wetback crisis' and an 'invasion.' More specifically, these reports blamed Mexican migrants for depressing wages in the Southwest, portrayed them as carriers of disease, accused them of contributing to deteriorating housing conditions across the interior of the country, and depicted them as a national threat.[4] At the same time, U.S. growers demanded continued access to a cheap and stable labor force. In response to growers' pressure, and after failing to convince the Mexican government to reopen recruitment centers in northern states, the U.S. government announced in January 1954 that it would begin issuing contracts to Mexicans who crossed the border.

At the same time, however, to address the xenophobic attitudes of the public and concerns about the rise of undocumented Mexican migration, in June 1954, the U.S. government launched Operation Wetback, an immigration enforcement operation to stem the flow of unauthorized (Mexican) migration through carrying out mass deportations. To reach its objective, the INS, in collaboration with state and local officials, launched a series of deportation drives along the U.S.-Mexico border and across cities including Los Angeles, San Francisco, Kansas City, and Chicago. According to a report from the INS, nearly 1 million apprehensions were made in 1954.[5] However, as historian Kelly Lytle-Hernandez (2010, 171) describes it, 'Operation Wetback was a larger than the usual deployment of the Border Patrol's familiar and failing tactics of migration control.' The amount of publicity that Operation Wetback received in the press triggered conditions of fear and uncertainty among Mexican communities, leaving many with

---

[4] See, 'Immigration: "Recalcitrant" Countries and the Use of Visa Sanctions to Encourage Cooperation with Alien Removals,' https://www.congress.gov/crs-product/IF11025

[5] Estimate calculated from the 'Termination of the Bracero Program: Some Effects on Farm Labor and Migrant Housing Needs (1965), U.S. Department of Agriculture: Economic Research Service, https://ideas.repec.org/p/ags/uerser/307309.html

limited options—including returning to their country of birth. Roughly 23,000 Mexicans were formally deported and 1 million left the U.S under 'voluntary departures'[6] in 1954. Deportation operations, however, continued throughout the 1950s. While the Mexican government has a long history of addressing discrimination against its diaspora through its network of consulates across U.S. jurisdictions, consular activities between the 1940s and 1960s were primarily focused on administering the Bracero Program. During this period, complaints about employer abuse and unilateral decisions regarding regulations and hiring practices were directed to U.S. authorities (Cano and Delano 2007). Mexican consulates did not have a clear mandate to protect unauthorized Mexicans living and working in the U.S. On the contrary, Mexican officials often sought support from U.S. authorities to deter unauthorized border crossings.

Although the Mexican government intended to use the Bracero Program to curb unauthorized emigration, the program triggered an influx of unauthorized migration to the U.S. In its early years, Mexico implemented measures such as heightened surveillance along its northern border, but these efforts proved insufficient—especially in light of the U.S. government's unilateral actions. While the Mexican government attempted to pressure U.S. officials, it ultimately viewed the Bracero Program as a tool to manage emigration more strategically. Rather than relying

---

[6] Estimates were retrieved from the 'Annual report of the Immigration and Naturalization Service United States Department of Justice, Washington D.C., Fiscal Year 1954'. See https://babel.hathitrust.org/cgi/pt?id=nnc1.cu08541337&seq=8. According to the U.S. Department of Justice's Executive Office for Immigration Review (EOIR), voluntary departures allow noncitizens to leave the U.S. at their own expenses within a specific period to avoid undergoing formal deportation order and its consequences including but not limited to lifetime bars from re-entering the U.S. Voluntary departures are granted by an immigration judge or officers for the U.S. Department of Homeland Security and are usually granted after an immigration agent apprehends a noncitizen. However, historian Adam Goodman (2020) argues that 'voluntary departures' is a government strategy to coerce noncitizens to leave the country. According to Goodman (2020), the U.S. has relied on voluntary departures as a cheaper mechanism to expedite the removal of noncitizens. See, *The Deportation Machine America's Long History of Expelling Immigrants.*

solely on border enforcement, Mexican officials engaged in dialogue with the INS and U.S.

Border Patrol to shape deportation strategies targeting unauthorized Mexican workers. Mexico's

role in crafting these policies became increasingly significant in the 1950s, as U.S. immigration

authorities framed the rise in Mexican apprehensions as an 'invasion,' intensifying calls for

enforcement and bilateral cooperation.

**Proactiveness in practice: Mexico's involvement in shaping deportation efforts from the U.S.**

*Enforcing its northern border*

On December 11, 1943, one year after the formal implementation of the Bracero Program, the

Mexican Embassy in Washington, D.C, sent a letter to the U.S. Department of State stating that

Mexico was experiencing a series of economic losses as a result of the 'surreptitious departure'

of its [agricultural] workers. In the letter, the Mexican government, stated that they had increased

vigilance at its northern border but that the country needed 'active collaboration' from the U.S.

government.[7]  The Mexican government also stated that 'the illegal border crossings under

reference must be a matter of concern for the two governments.' Mexico was requesting the U.S.

government to secure its borders, otherwise it would see itself having to revise the joint labor

agreements. One week later, Chief J.F McGurk assistant to Earl G. Harrison the Commissioner

of INS explained to Harrison that the INS, needed to address Mexico's demands of extending its

'vigilance at the border to prevent clandestine and illegal entry of Mexican workers into the

---

[7] Letter from Mexican Embassy Washington, D.C, December 11, 1943, file: 56161/109, Box 19133, Research Group 89, NARA.

U.S.'[8] Such exchange made it evident that during the earlier stages of the Bracero Program, the

U.S. had higher levels of vulnerability. On the one hand, the country needed access to cheap

labor to address the labor shortages and knew that they needed Mexico's assistance in managing

cross border migration flows.

The Mexican government adopted measures to enforce its northern border to address the

economic and political implications of a mass [unauthorized] exodus. The Mexican government

used different state actors, from commissioning military personnel and law enforcement officials

to police the border.[9] During the early years of the Bracero Program, the U.S. Border Patrol

dropped off deportees at the Mexican border, making it easy for deportees to cross back to the

U.S. To address this concern, in 1945, President Ávila Camacho (1940 to 1946) requested the

Ministry of Finance and Public Credit (SHCP) to disburse the amount of 20,600 (MXN) to

transport deportees from the northern state of Sonora to the central state of Jalisco via trains.[10]

1945 marked the year the Mexican government adopted measures to transport Mexicans

from the border into the interior of the country against their will and at the government's

expense. The purpose of these early modes of transportation from the border into the interior of

Mexico was to punish and deter deportees from re-emigrating to the U.S.[11]

---

[8] Response from J.F. McGurk, Assist Chief to Earl G. Harrison, Commissioner, INS, Department of Justice, Philadelphia, PA, December 22, 1943, file: 56161/109, Box 19133, Research Group 89, NARA.
[9] Response letter from President Manuel Ávila Camacho to the Ministry of National Defense, February 1944, Mexico City, (AGN), Mexico, File Manuel Ávila Camacho 56161/109, Box, 793.
[10] Memo from Mexican President Manuel Ávila Camacho (1940 to 1946) to the Ministry of Finance and Public Credit (SHCP) requesting disbursement of 20,600 (MXN) to rail companies to transport Mexican deportees, February 1945, Mexico City, Archivo General de la Nación (AGN). File Manuel Ávila Camacho 56161/109, Box, 793.
[11] It is important to note that the 1917 Mexican Constitution guaranteed Mexican citizens the right to enter and exit national territory. However, from the 1920s through the termination of the Bracero Program, the federal government implemented a series of 'instruments of emigration control' (Fitzgerald 2006). As Fitzgerald (2006) argues, the government's efforts to regulate unauthorized emigration were shaped by

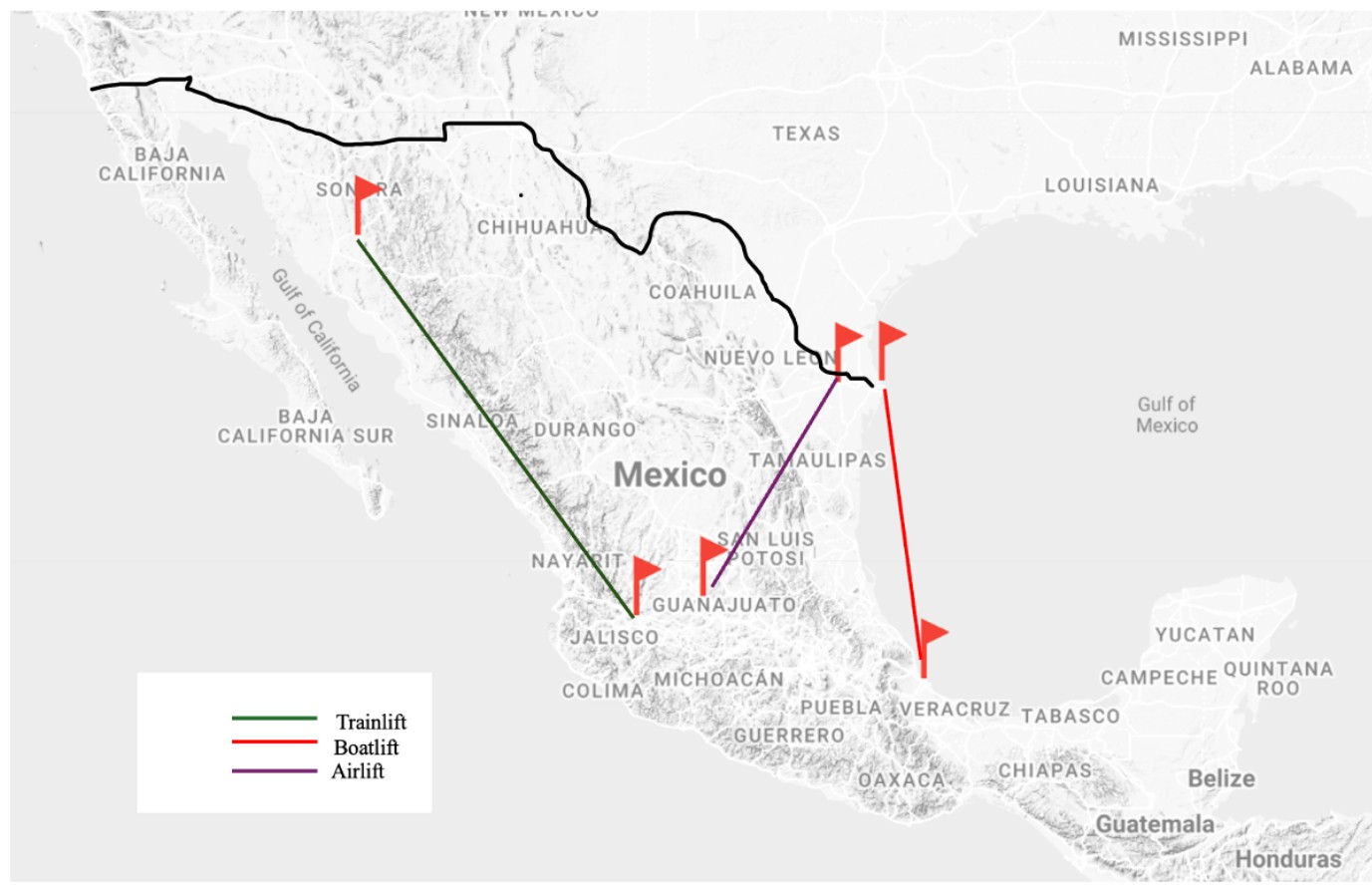

Figure 1. Map of routes to transport Mexicans into the interior of the country via train lifts, boatlifts and airlifts. [Map elaborated by author.]

*The Mexican boat lifts from 1954 to 1956*

Apart from the train lifts, between 1954 to 1956, the U.S. government used vessels to transport

Mexican deportees from Port Isabel, Texas, to Veracruz, Mexico. The vessels used to transport

Mexican deportees were owned by Mexican transportation companies, including the Transportes

---

domestic factors—such as the demands and interests of state and municipal actors—and international factors, including the U.S. demand for a cheap and flexible labor force. The Bracero Program was significant for Mexican federal authorities, as it served as a mechanism to manage who could legally participate in temporary labor migration. Nevertheless, scholars have noted that the Mexican state consistently failed to control unauthorized emigration to the U.S.

Marítimos Refrigerados S.A (TMR) and the Transportes Marítimos y Fluviales (TMF). The U.S. government signed a series of contracts with the TMR and the TMF because these companies offered the cheapest option for removing Mexicans from U.S. territory. The cost of renting a vessel for each voyage ranged from 72,000 to 90,000 Mexican pesos, and the price for transporting a deportee was 8 U.S dollars.[12] Roughly 800 deportees were transported in each voyage, including men, women, and children.[13] The first vessel departed port Isabel on September 3, 1954.[14]

Although the Mexican government was not involved in negotiating transportation contracts with the U.S. government, nor did they directly profit from the contracts, based on informal agreements with officials from the INS including letters and copies of contracts, the Mexican government was responsible for inspecting the vessels and approving the use of vessels – necessary steps for carrying out deportations from the U.S. Furthermore, the Mexican government was also responsible for creating an infrastructure at the port of Veracruz for processing Mexican deportees and for transporting them into the interior of the country. The process for carrying out deportations via boatlifts required close engagement between U.S. and Mexican authorities. A few days before the scheduled date of the deportation, Mexican and U.S. officials, including staff from the Consulate of Brownsville, Texas, officials from the Secretariat of the Interior (SEGOB), and officials from the INS were in charge of inspecting the vessels and

---

[12] Telegram, from TMR to INS sharing the cost of each passenger. November 10, 1955, file 56364/043.36, box 21936, Research Group 85, National Archives and Records Administration, Washington, D.C. (NARA)

[13] More than fifty U.S. Border Patrol reports mention that 800 deportees were inside the vessels. See file 56364/043.36, box 21936, Research Group 85 (NARA)

[14] Congressional report on the vessels that the U.S. government used to deport Mexican noncitizens presented by William P. Rogers, U.S Deputy Attorney General, August 23, 1956, file 56364/043.36, box 21936, Research Group 85, (NARA)

sharing a list of the deportees that would go onboard.[15] On the day of the deportation, women and children were placed in the same deck, while the men were separated. Each voyage lasted approximately forty to sixty hours.[16] Once the vessel arrived at the port of Veracruz, Mexican immigration officials escorted deportees to provisional warehouses where they remained under the custody of the SEGOB and Mexico's immigration officials. Individuals who were deported for the first time were released or were transported to Mexico City via trains or buses. According to reports from Border Patrol officers, 'repeaters'—individuals who had been deported more than once—were handled separately and transferred to the Allende prison in Veracruz, where they received a 15-day jail sentence as punishment to deter them from re-emigrating to the U.S.

---

[15] Ibid.,
[16] Ibid.,

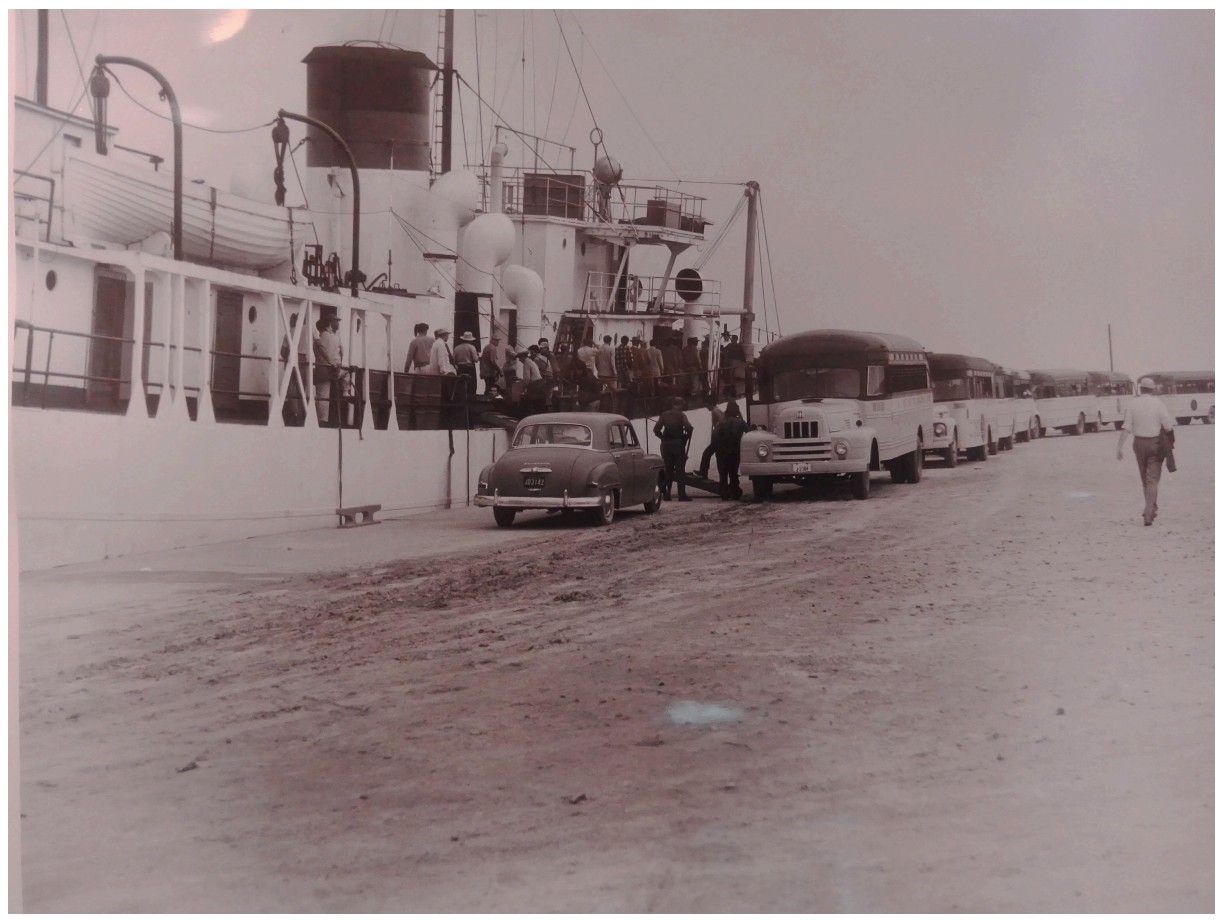

Figure 2. Mexican deportees loading onto the *Mercucio*, August 8, 1956, file: 56364/043.36, Research Group 83, Box 21936, National Archives, Washington, D.C.

Not much is known about the experiences that the roughly 49,000 deportees encountered during the deportation boat lifts. However, an incident inside one of the vessels in 1956 generated media coverage, revealing the conditions deportees were exposed to. On August 28, 1956, the *Mercurio* one of the vessels owned by the Mexican contract company TMR, had to make an emergency stop at the Port of Tampico in the northern Mexican state of Tamaulipas. While the *Mercurio* was in the middle of the Pánuco River awaiting to cast anchor, over 500 of

the deportees 'staged a riot' inside the vessel.[17]According to reports, thirty-six passengers

jumped, five drowned and only three bodies were recovered.[18] The *Mercurio* incident led to U.S.

and Mexican media coverage and independent investigations by Mexican and U.S. authorities.

Details about the precarious conditions deportees encountered led to public outrage by Mexico's

opposition, local governments, and religious organizations. Media reports revealed that the

*Mercurio* was designed to carry cargo and was not equipped for passengers.[19] The *Mercurio*

lacked proper facilities to transport passengers safely, details that the Mexican government was

aware of during their inspection routines but ignored. The vessel, for example, lacked sufficient

lifejackets for the 800 passengers on board and lacked sleeping facilities.[20] In the U.S., after

requesting a Congressional inquiry, U.S. Representative Robert H. Mollohan from the state of

West Virginia described the *Mercurio* as a 'hell ship' with 'deplorable conditions.'[21] In

interviews, deportees stated that some jumped off the vessel because they were from northern

Mexico and did not want to be transported south to Veracruz as this would require additional

funds to return to their hometowns—funds that the Mexican government was unwilling to offer

to deportees.[22] High-ranking Mexican federal authorities from the SEGOB and the SRE blamed

the deportees and framed them as criminals, stating that the incident was politically inspired

---

[17] Ibid.,

[18] Telegram from Mexico City to the U.S. Department of State stating three bodies were recovered. August 29, 1956, file 56364/043.36, box 21936, Research Group 85, (NARA)

[19] Telegram from TMR to INS listing details and prices for renting the vessel. In the telegram, representatives from TMR state that the vessel transports cargo including bananas from Veracruz to the U.S., November 17, 1955, file 56364/043.36, box 21936, Research Group 85, (NARA)

[20] Newspaper clipping, 'Rep. Mollohan Orders Probe of 'Hell Ship,' Washington News, August 28, 1956, file 56364/043.36, box 21936, Research Group 85, (NARA)

[21] Ibid.,

[22] Ibid.,

'aimed at embarrassing' Mexico's presidential administration.[23] However, based on letters written by deportees and interviews with the press, most of the deportees inside the *Mercurio* were individuals caught attempting to cross the border or who had lived for extended periods in the U.S., and were not affiliated with Mexican political parties. On September 7, 1956, the SRE suspended all boatlifts from Texas to Veracruz. Although the Mexican government was forced to cancel all boatlifts out of the public scrutiny they received in the U.S. and Mexican press, the government continued to proactively facilitate the deportation of its citizens through removal procedures less visible to the public.

*The Mexican Airlifts*

Following the incident of the *Mercurio*, the Mexican government adopted different measures for facilitating the deportation of unauthorized Mexicans into the interior of the country, this time through airlifts. On November 10, 1957, an airplane departed from the Mexican northern border town of Reynosa, Tamaulipas, to the Mexican city of León, located in the central state of Guanajuato.[24] According to INS reports, about sixty passengers were onboard the airplane. Deporting Mexicans from the U.S. to León required coordination between U.S. immigration officials and local authorities in Mexico. In the U.S., deportees were transported from detention centers to Hildalgo, Texas. Once deportees were in Hidalgo, they were driven by U.S. Border

---

[23] Newspaper clipping, 'Barco, Contracto, Maltrato, Hambre: Todo Es Nusestro! La Historia Patriotera Tira Piedras Sobre Nuestro Propio Tejado Indefenso' Zocalo, August 27, 1956, file 56364/043.36, box 21936, Research Group 85, (NARA)

[24] Copy of agreement between the U.S. and Mexico regarding airlifts from Reynosa to Leon. The report provides details about the Mexican actors and institutions involved in overseeing the flights, the dates the flights took place and how Mexican deportees were to be processed once they entered Mexican territory (in Reynosa, January 14, 1960, file: 56364/043NW to 56364/043 box 13, Research Group 85, (NARA)

Patrol officers or were forced to cross (by foot) the Reynosa-Hidalgo International Bridge. On the Mexican side, deportees were processed by Mexican immigration officials and transported to board planes to the Mexican city of León in the state of Guanajuato.[25] The logic behind transporting Mexicans against their will from Mexico's northern border into the interior of the country was the same as the boatlifts, to punish and deter deportees from re-entering the U.S. According to a report by an INS attaché in Mexico City, once deportees arrived in León, they were transported to railroad stations and were only provided transportation to locations south of León,[26] leaving those from northern Mexico stranded with limited financial resources.

It is important to note, however, that airlifts were not a new method for transporting Mexican deportees. According to historian Adam Goodman (2020), the first airlift occurred in 1946, but these airlifts only transported deportees within U.S. territory from Arizona to Texas to then be deported by foot or bus to the Mexican side of the border. In 1951, the U.S. government used private cargo airlines to transport Mexican deportees to various Mexican cities including, Durango, San Luis Potosi, and Guadalajara, but these airlifts were suspended in 1952 after INS failed to secure funding from Congress.[27] However, what made the airlifts from 1957 to the late 1960s different than previous years was the level of engagement from the Mexican government in leading these deportation efforts. The Mexican government contracted Mexican airlines to transport Mexicans to León.

---

[25] Summary of call between Charles and Gustavo Diaz Ordaz regarding procedures to carry out airlifts. November 6, 1957, file 56364/043NW to 56364/043, box 13 Research Group 85, (NARA)
[26] Ibid.,
[27] Annual report of the Immigration and Naturalization Service, June 30, 1952, Department of Justice. See section Border Patrol, page 41, 'Airlifts.'
https://babel.hathitrust.org/cgi/pt?id=mdp.39015004046655&seq=59

Furthermore, the Mexican government funded all the airlifts to transport deportees into the interior of the country. The Mexican government used its national airline Líneas Áreas Unidas S.A (LAUSA) to transport Mexicans from Reynosa to León.[28] The Mexican government in dialogue with U.S. government officials agreed to use its national airlines and airlifts as they considered it a faster method for transporting deportees into the interior of the country. During the negotiation process of implementing airlifts to Mexico, the INS wanted Mexico City to be the final destination for deportees. However, the Mexican government was hesitant as Mexican officials described Mexico City as the 'hub of the Mexican press.'[29] The Mexican government was concerned that the airlifts to Mexico City, along with the conditions under which deportees were transported, would provoke heightened scrutiny from the press. In the context of the upcoming 1958 presidential elections, officials sought to preempt criticism that might undermine the legitimacy or public image of the ruling Institutional Revolutionary Party (PRI).[30] The Mexican government selected León, Guanajuato as the processing site for deportees due to the support offered by municipal and state officials, who viewed the deportation flights as an opportunity to generate local employment and profit. The mayor of León, for example, stated that the 'airlifts did not cause usual problems to the city' and that he was 'grateful for the small amount of business' the airlifts were bringing to his city.[31]

---

[28] Report issued to all Border Patrol Inspectors, Southwest Region describing personnel and airline in charge of transporting Mexico to Leon via airlifts. November 5, 1957, file 56364/043NW to 56364/043, box 13, Research Group 85, (NARA)

[29] Ibid.,

[30] Ibid.,

[31] Report from INS commissioner summarizing the responses from state officials from Leon, Guanajuato regarding airlifts. January 30, 1958, file 56364/043NW to 56364/043, box 13, Research Group 85, (NARA)

**Unpacking Mexico's Interests**

During the early phases of the Bracero Program (1942–1964), the Mexican government not only adopted various measures to control unauthorized emigration flows but also became a proactive actor in the U.S.'s deportation efforts. Mexican federal authorities pressured U.S. officials to regulate unauthorized border crossings and implemented mechanisms to process Mexican nationals deported to border cities in Mexico. This proactiveness in deportation efforts intensified during the mid-1950s through the 1960s. The Mexican government took a more active role by designing strategies and approving procedures for how deportations should be carried out. In dialogue and collaboration with U.S. immigration authorities, Mexico developed punitive measures focused on transporting deportees in inhumane conditions, relocating them to the country's interior—far from their places of origin. This increased involvement corresponded in moments when U.S. federal authorities framed the rise in border apprehensions as an 'invasion,' when the INS lacked funding to conduct aerial deportations, and when powerful agricultural interest groups pressured Mexican officials to prevent laborers from emigrating to the U.S. While the previous section examined how Mexico became a proactive actor in U.S. deportation proceedings, this section explains why Mexican officials chose to take on such a role despite the potential for political and economic backlash.

After President Lázaro Cárdenas (1936–1940) left office, Mexico shifted from supporting agrarian reform to prioritizing policies focused on expanding large-scale industrialization (King 1970). To promote agricultural industrialization, President Miguel Alemán Valdés (1940–1946) invested in irrigation systems, dam projects, road infrastructure, and agricultural research as part

of the Green Revolution, aimed at enhancing agricultural productivity (Henderson 2010). These public investments, however, concentrated in northern Mexico—a region where profitable crops such as cotton had thrived since the late 19th century and were deeply tied to the country's political economy (Walsh 2000). Between 1941 and 1952, over ninety percent of Mexico's agricultural budget was allocated to irrigation projects in the northern states (Henderson 2010, 60). Moreover, these investments disproportionately benefited commercially oriented agribusinesses (King 1970, 33), while only minimal support was directed to *ejidatarios,* or members of communal agricultural lands. Due to the growing importance of these commercially oriented private farmers, the Mexican government became increasingly dependent on their production, and such groups gradually gained greater influence in Mexican politics.

With the conditions set in place, including the infrastructure and political base to increase the exportation of products, mainly cotton, private farmers needed a cheap and flexible labor force. Following the implementation of the Bracero Program, a series of interest groups, from agribusinesses to industrial associations, began putting pressure on President Alemán Valdés to deter the flow of unauthorized labor migration. Interest groups described the exodus as a 'strain on agricultural production'[32] and stated that the lack of workers was rotting away cotton harvest.[33] To address the grievances of interest groups, the Mexican government not only enforced its northern border by sending military and immigration officials as well as requesting assistance from the U.S government in surveilling its southern border to curtail emigration, but

[32] Letter from Coronel Gabino Vizcarra the President of la Legion Mexicana to President Manuel Ávila Camacho, July 16, 1943, AGN, Mexico City.
[33] Letter from the National Chamber of the Transformation Industry to President Ávila Camacho, December 20, 1945, AGN, Mexico City.

the Mexican government also became involved by designing strategies to deport its citizens away from the U.S.-Mexico border. One of those strategies, as discussed, featured deporting deportees into the country's interior. The purpose of this deportation strategy was to punish and deter deportees from re-emigrating to the U.S, hoping to control the mobility of Mexican agricultural laborers.

The Mexican government also became increasingly proactive in facilitating the deportation of its citizens as a way to strengthen its diplomatic relationship with the U.S and to secure the benefits associated with that relationship—including the continuation of the Bracero Program and support on other issues such as commercial trade. While the rise in unauthorized labor emigration was a growing concern for the Mexican government, it was not initially a significant issue for the U.S. government during the early stages of the Bracero Program. Throughout the program's duration, the U.S. government often took unilateral actions that conflicted with Mexico's interests, such as opening the border to unauthorized labor migration (Calavita 1996; Craig 1971). However, unauthorized Mexican migration became a more pressing concern for U.S. officials in the mid-1950s, particularly as public and media pressure increased. At that time, unauthorized Mexican migrants were increasingly blamed for rising unemployment and for the social ills the region was experiencing.

The U.S. government sought to address the so-called 'wetback invasion' by increasing the presence of the Border Patrol along its southern border and launching a massive 'round-up of unauthorized migrants'—an immigration enforcement campaign known as Operation Wetback. However, to make its immigration enforcement policies effective, the U.S. government needed

Mexico's assistance. Before launching Operation Wetback, in 1952, the Department of Justice (DOJ)—the agency responsible for overseeing immigration enforcement—requested funding from the House Appropriations Committee to construct a 150-mile fence along the border, but the request was denied. The DOJ also requested funding to transport Mexican deportees into the interior of Mexico via U.S. airlifts. This request was also denied,[34] which meant that the Immigration and Naturalization Service (INS) lacked the means to relocate deportees away from the U.S.-Mexico border as intended.

The Mexican government saw the INS's vulnerability in carrying out its border and deportation efforts as an opportunity to assist the U.S. Mexico's proactiveness became most evident during the summer of 1954, when the INS launched Operation Wetback and in its aftermath. In addition to accepting deportees at the border, the Mexican government encouraged and facilitated the deportation of its citizens by approving contracts between privately owned Mexican boat, aerial, and bus companies and the U.S. government to transport deportees from the border into Mexico's interior. These actions allowed the Mexican government not only to address the outflow of unauthorized labor migrants—responding to pressure from northern Mexican economic elites—but also to support the continuation of the Bracero Program. Following Operation Wetback, the INS increased the number of Bracero contracts, one of Mexico's key demands. The number of braceros admitted rose from 309,033 in 1954 to 445,197 in 1956.[35]

---

[34] Annual Report of the Immigration and Naturalization Service for the Fiscal Year 1952. Department of Justice. See page, 41 subsections 'The Airlifts.'
https://babel.hathitrust.org/cgi/pt?id=mdp.39015004046655&seq=59
[35] Numbers retrieved from Table 1. 'Foreign workers admitted for temporary employment in U.S. agriculture, by year and nationality' in report Termination of the Bracero Program: Some Effects on Farm Labor and Migrant Housing, U.S. Department of Agriculture: Economic Research Service, 1965.
https://ideas.repec.org/p/ags/uerser/307309.html

For the Mexican government, the continuation of the Bracero Program was essential, as it helped alleviate domestic challenges—most notably, unemployment, which disproportionately concentrated in central-western and southern Mexico. Although Mexico's industrialization policies in the mid-twentieth century stimulated economic growth and increased production, the benefits were unevenly distributed, especially in the agricultural sector (Moreno-Bird and Ros 2009, 103). Between 1940 and 1965, crop production grew at an average annual rate of 5.7% (Moreno-Bird and Ros 2009, 103), but agricultural investments largely favored export-oriented private growers in northern Mexico. In contrast, the central-western regions, where agriculture was primarily organized around *ejidos,* were neglected.[36] Ejidos were communal land grants in which recipients could cultivate and sell crops, either individually or collectively, but could not sell the land itself.[37] President Lázaro Cárdenas (1934–1940) is credited with implementing the most radical land reforms in Mexico's history. Between 1935 and 1940, more land was distributed as *ejidos* than under any previous administration (Levitsky and Way 2022, 147). During his presidency, Cárdenas distributed approximately 50 million acres of land to 811,157 individuals (Dwyer 2008, 82). However, his successors significantly slowed down the pace of land redistribution.

---

[36] An ejido is a system of collective land that the Mexican government expropriated from wealthy landowners and distributed to local groups of peasants mainly for agricultural purposes.

[37] Although the concept of ejidos is rooted in pre-revolutionary indigenous community models, the ejido system established under Article 27 of the 1917 Mexican Constitution allowed the government to expropriate land from large landowners and distribute it to rural communities. Members of ejidos, or ejidatarios, were granted the right to use portions of the land mainly for agriculture, but they did not have the right to sell or lease the land. To request land in the form of an ejido, as Sellars (2019, 1217) describes, required 'citizen cooperation.' Individuals interested in requesting government sponsored land had to file a petition as a group to their state governor describing why they needed land in their villages (Albertus et. al 2015). Once a petition was filed, it was up to state and federal officials to review the petitions and locate portions of land that could be expropriated to form ejidos (Sellars 2015).

Out of fear that the *ejido* system would slow down agricultural investment, in 1940, the Mexican government under newly elected President Manuel Ávila Camacho (1940 to 1946) adopted a series of legal provisions designed to slow down land distribution to peasants. In 1940, the administration provided more legal protections to private landowners from land expropriation, and in 1942, the Mexican government reduced the number of hectares of land that could be expropriated (Albertus et al., 2016). Furthermore, unfavorable weather conditions and population growth left many individuals without access to land, motivating many to seek employment in the U.S. through the Bracero Program. Hence, the Mexican federal government needed access to more bracero contracts to meet the demands of state and local officials and to prevent men from emigrating without authorization.

By assisting the U.S. in carrying out its deportation efforts during the 1940s to the 1960s, the Mexican government was able to secure additional Bracero contracts. Following Operation Wetback**,** the U.S. Immigration and Naturalization Service (INS) expanded the number of Bracero contracts—an outcome that served both Mexico's interest in addressing unemployment and U.S. growers' demand for labor. The number of admitted braceros rose from 309,033 in 1954 to 445,197 in 1956,[38] while border apprehensions and deportations declined during the same period. For Mexico, additional Bracero contracts helped mitigate unemployment and reduce the risk of collective unrest in regions with high rates of unemployment (Sellars 2015). Braceros were crucial for Mexico's political economy as they stimulated rural development

---

[38] Numbers retrieved from Table 1. 'Foreign workers admitted for temporary employment in U.S. agriculture, by year and nationality' in report Termination of the Bracero Program: Some Effects on Farm Labor and Migrant Housing, U.S. Department of Agriculture: Economic Research Service, 1965. https://ideas.repec.org/p/ags/uerser/307309.html

through the remittances they sent to their communities. By the 1950s, these remittances had become Mexico's third-largest source of hard currency (Cohen 2011, 24).

**Discussion and Conclusion**

How do origin countries respond to the deportation efforts designed by host countries? Scholars have highlighted that for deportations to be carried out successfully, host countries depend on inter-state cooperation from origin countries (Ellerman 2008; Cassarino 2007). Host countries have adopted strategies, such as bilateral agreements, informal schemes, threats, and concessions to secure inter-state cooperation in readmission processes. Origin countries, however, are not passive actors that will always accept the demands of countries seeking to deport noncitizens.

Although this study examined events from the 20[th] century, 'recovering historical depth' (Tolay 2023) at the state archives in the U.S. and Mexico, provides valuable insights for understanding contemporary processes on deportation and what is at stake for origin countries. First, by introducing the concept of proactiveness, this paper suggests that origin countries may engage proactively not only in response to external pressures from host countries, but also as a strategic move to secure both material and intangible gains—such as strengthening diplomatic ties and fostering relations with nonstate actors, including but not limited to for-profit entities. The likelihood of proactive engagement by origin countries depends, in part, on the vulnerability of host countries—particularly during periods when host countries are unable to carry out deportation and border policies independently and must rely on origin countries. In such contexts, origin countries will have more leverage to shape the terms of cooperation, thereby creating opportunities to demand resources or concessions that align with their political objectives, such as trade privileges, relaxed travel restrictions for their citizens, access to loans or

aid. At the same time, this study indicates that origin countries are less inclined to adopt a proactive role when deportation does not serve their political objectives. Moreover, when origin countries already receive concessions or resources from host countries through other policy channels, they may have fewer incentives to become proactive actors in deportation efforts.

Although the number of deportations reached its highest peak between 2009 to 2016,[39] the Mexican government has adopted passive responses to manage the deportation of its citizens from the U.S. The Mexican government has not much to gain from being a proactive actor in the deportation efforts from the U.S. Domestically, since the 1980s, controlling unauthorized emigration is no longer a priority for the Mexican government (see Minian 2018). In terms of its foreign policy, the Mexican government has been able to solidify its diplomatic relationship with the U.S. government through trade agreements, and border and security issues. While Mexican immigration authorities process the deportation of its nationals, Mexico is no longer involved in providing the U.S. government with modes of transporting deportees nor has it adopted a discourse and strategies that explicitly promote deportation. While the Mexican government has not been a proactive actor in facilitating the deportation of its nationals, Mexico has been proactive in other migration related issues such as deterring flows of migrants from reaching the U.S.-Mexico border.

Beginning in the 1980s, the Mexican government has adopted a series of incentives designed in dialogue with U.S. authorities on how to deter non-Mexicans from transiting through Mexico.

---

[39] Roughly 2.2 million Mexicans were formally removed from the U.S. to Mexico. Estimate was calculated using data from the 2018 Yearbook of Immigration Statistics https://ohss.dhs.gov/topics/immigration/yearbook/2018. Formal deportations are officially categorized as removals by U.S. immigration authorities.

Throughout the years, the Mexican government has mobilized military and police presence in Mexican southern border towns, established roadblocks and mobile checkpoints across highways from southern to east-central Mexico, constructed detention centers and carried out raids and arrests (Chavez and Voisine 2019). By supporting the U.S. government in deterring flows of migrants from reaching the U.S.-Mexico border, the Mexican government has accessed resources that have important domestic implications. With financial assistance from the U.S. government, Mexico has, for example, been able to use U.S. funding to expand military training, combat criminal organizations, and stimulate development in southern Mexico—one of the poorest regions in the country.[40]

As deportation becomes a salient mechanism that governments across the global north and global south use not only to manage unwanted migration flows but also to attract specific sectors of the electorate, we need more research on how origin countries respond to the rhetoric and policies of host countries regarding deportations. More specifically, we need more comparative research that explores how factors not discussed in this paper shape level of proactiveness including but not limited to regime type as well as colonial and historical legacies. Additionally, further study is warranted on the conditions under which countries are willing to facilitate the deportation of individuals who are not their own nationals. A striking example is President Nayib Bukele's proposal to accept deportees of any nationality into El Salvador's Terrorism Confinement Center

---

[40] In 2021, the U.S. and Mexican government announced the launched of Sembrando Oportunidades, to address the root causes of irregular migration from Mexico and Central America. The agencies responsible for implementing the program include the United States Agency for International Development (USAID) and the Mexican Agency for International Development Cooperation (AMEXCID). See, 'FACT SHEET: 2023 U.S. Mexico High-Level Economic Dialogue (Pillar II),' Office of the United States Trade Representative. https://ustr.gov/about-us/policy-offices/press-office/factsheets/2023/september/fact-sheet-2023-us-mexico-high-level-economic-dialogue

(CECOT), which was carried out in March 2025 with the transfer of more than 230 Venezuelan citizens. Ultimately, this paper aims to contribute to broader debates on how and why governments increasingly use deportation as a mechanism to pursue political agendas and what are the implications of this practice.

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

**Appendix**

Achieves Consulted

Archivo General de la Nación (AGN), Mexico City, Mexico

> o Fondo Manuel Avila Camacho, file 546.6/120, box, 793

National Archives and Records Administration, (NARA) Washington, D.C

> o File 659.4, research group 85, box 21916
> o File UD-05W8, research group, box 13
> o File 56161/109, research group, box 19133
> o File 56364/043.36Pt.1 to 56364/043.36 Pt 1. research group 85, box 21936

Online reports from the Immigration and Naturalization Service Annual Reports

> o Annual report of the Immigration and Naturalization Service, 1954 U.S. Department of Justice, Washington D.C.: https://babel.hathitrust.org/cgi/pt?id=nnc1.cu08541337&seq=8
> o Annual report of the Immigration and Naturalization Service, 1952, U.S. Department of Justice. https://babel.hathitrust.org/cgi/pt?id=mdp.39015004046655&seq=59