# Peer review of "Beyond Cooperation: The Role of Origin Countries in Deportation Efforts, Evidence from Mexico (1942 to 1964)"

_Migration Politics_

## Round 1 · Referee Report · Anonymous (Referee 1) · 2025-1-14

Strengths

This is a terrific paper. I truly enjoyed it. I would recommend the author to read page 109 of Acosta the National versus the foreigner (2018, CUP) where the author argues that "Latin Americans strategically banded together to oppose any possible restrictions that the USA may have suggested against their own nationals" in the early 20th century. This is also analysed by Fitzgerald and Cook Martin in their 2014 book culling the masses, around p. 76. I wonder whether the author would be able to explain a bit further in one or two paragraphs how Mexico, and to what extent, changed a previous position opposing discrimination of their own nationals in the US. I think the author clearly explains the motivations and concerns about the bracero program and also how businesses in Mexico needed workers, but again it would be interesting to see a bit more, even if very briefly, about how this changed a previous political position in the past or not.
As a lawyer, I find it interesting that Mexico supported the Universal Declaration of Human Rights where, one of the rights enshrined is the right to leave your own country. This might be outside the scope of the paper but was there any discussion at legal level about how not allowing nationals to leave the Mexican territory breached this fundamental right?
Is there anything to say in the conclusion about how the past shows us lessons, if any, for the current situation, notably now that we have two new presidents in both countries and, possibly, a difficult future relationship regarding expulsions?

Report

Please see above.

Recommendation

Publish (easily meets expectations and criteria for this Journal; among top 50%)

---

## Round 1 · Referee Report · Anonymous (Referee 2) · 2025-2-25

Strengths

This paper explains very convincingly, drawing on a wide variety of sources, how the Mexican government became an actor in facilitating the deportation of its own citizens from the 1940s to the 1960s. It highlights the means used by the Mexican state to facilitate the deportation of its citizens: the government chartered ships and airplanes to take part in the deportation of its own citizens expelled from the United States.

In a broader sense, this episode, well analyzed by the author, allows us to reconsider the role of the origin countries of the deportees, which even today actively support these procedures by participating in the return of deportees to their borders and by using violence against them.

Weaknesses

This reviewers considers a few weaknesses that should be addressed in a revised version of the article: 1. the methodology should be shortened and not presented as part of the argument, but rather instrumental to the demonstration 2. the initial section lacks historical contextualisation 3. the second section requires clarification on the nature of sources (archives) and their treatment - quantitative qualitative (this can go in the introduction) 4. emphasise and detail the contradiction in Mexican policies 5. offer a segmented periodisation of the historical sequence analysed to identify turning points, continuities and various changes in observed dynamics

Report

From the point of view of the organization of the plan of the article, as presented in the introduction, it is somewhat surprising that the author devotes a central part of the argument to the methodology, which should be explained right from the introduction and applied throughout the development of the article.

After the introduction, the first part of the article begins with an examination of the relationship between countries of origin and host countries, showing that today it can be characterized by the reluctance of countries of origin to readmit their nationals. Nevertheless, it may be useful to first provide some context for the historical period under study before moving on to the present.

In the second section dealing with methodology, the nature of the archival research carried out for this study needs to be clarified: obviously, the author was not able to consult all the federal archives relating to Mexican deportees from the US during that time period, and it would have been interesting to know how the author selected the documents, letters, reports and articles mentioned at the end of p. 9.

Similarly, the fact that the author consulted a great many archival documents (see p. 10 "I examined over six hundred pieces of archival data") does not guarantee that she carried out quantitative research guided by the investigative methods of quantitative history. The many documents mentioned remain quite vague. With regard to the sources that the author uses here, one could wonder about the archives left by the deportees themselves, or about the advisability of conducting oral interviews with some of those who experienced these harsh conditions of repatriation to Mexico.

The heart of the article's argument is found on page 16 and following: the author shows the economic dimension of Mexico's "proactiveness" in reclaiming its citizens who had crossed illegally into the United States. In December 1943, the Mexican Embassy in Washington D.C. reported that these illegal departures were causing the country an economic loss. This explanation of Mexico's proactive policy, which is developed later on in the article (p. 26, discussing the lack of Mexican labourers for the 1945 cotton harvest), should perhaps be considered at the outset. Nevertheless, this same proactive policy of repatriating illegal Mexicans had a cost (assessed on p. 19), and it would therefore be interesting to examine the contradictions of the Mexican policy, aimed at recovering the nation's vital forces, but costly in other respects.

Finally, I wonder about the period covered, which is relatively long (two decades): apart from the changes in the means of transportation, which are well analyzed, it would have been interesting to take better account of the changes observed during this rather long period. Was Mexican policy always so proactive throughout the whole period? Were there fluctuations due to economic and political conditions or diplomatic relations? Perhaps the paper, which is already very convincing, could pay more attention to these variations over time.

Requested changes

Requested changes indicated in the weakness section.

Recommendation

Ask for minor revision

---

## Round 2 · Referee Report · Anonymous (Referee 1) · 2025-9-2

Strengths

I am happy for the paper to be published with the changes made.

Report

I am happy for the paper to be published with the changes made.

Recommendation

Publish (easily meets expectations and criteria for this Journal; among top 50%)

---

## Round 2 · Referee Report · Delphine Diaz (Referee 3) · 2025-9-30

Strengths

I appreciate the revisions made, and I find that the author has genuinely sought to address our questions and comments.

Weaknesses

That said, it seems to me that she could still provide more detail on the archival series consulted (NARA on the U.S. side and AGM on the Mexican side). On p. 9, the choice of collections consulted could be explained more explicitly. But this may simply reflect my historian's perspective, whereas Guadalupe Chavez comes from another discipline.

The photograph inserted on p. 21 would benefit from further commentary.

Report

These are minor suggestions, and in my view, they do not require further validation by this reviewer.

Requested changes

  • The photograph inserted on p. 21 has to be commented in the text.
  • A small detail: on p. 20, in the notes, "Ibid." should be followed by a full stop, not a semicolon after the stop.

Recommendation

Publish (surpasses expectations and criteria for this Journal; among top 10%)

---

## Round 2 · Author Response

I want to thank the reviewers for taking the time to engage with my work and for providing valuable feedback. I have carefully addressed their comments and suggestions. In particular, as suggested by Reviewer 2, I shortened the methodology section and specified the nature of the sources I collected and how I selected those used for this paper. Second, I provided more context about the period the paper examines and why it was a significant period in Mexico. This information can be found in the introduction. Third, in line with Reviewers 1 and 2, I address what the broader public can learn from this case study and the value it adds to understanding contemporary deportation processes from the perspective of origin countries. Lastly, to address the editor’s comments, the revised version discusses how the paper contributes to the field of migration diplomacy through its rich content and by introducing the concept of proactiveness. I have also created a separate document in which I directly address the questions and comments provided by the reviewers.

---

## Round 2 · List of Changes

Introduction
- I added a new paragraph in the introduction with context on why the 1940s to the 1960s was a period of rapid transformation in Mexico. New paragraph can be found on page 2
- I expanded the definition of proactiveness
- On page 4 I discuss how the paper contributes to the field of migration diplomacy

Methodology section
-As suggested by Reviewer 2, this section was shortened. The section does not focus on how much data was analyzed, but instead, the section describes the nature of the sources collected, how I selected the archival sources, and the type of sources I used for this paper.

The Context: The Bracero Program (1942 to 1964) and managing cross-border mobility
- As suggested by Reviewer 1 in this section, I added a few sentences to address how Mexican consulates responded to the discrimination of Mexican nationals during the 1940s to the 1960s. This update can be found on page 15.

Unpacking Mexico’s Interests
- As recommended by Reviewer 2, I added a few sentences describing how Mexico’s responses varied throughout the 1940s to the 1960s. I mention that Mexico’s degree of proactiveness intensified during the 1950s and explain the conditions that led to such change. These changes can be found on page 26.

Discussion and conclusion

- This section explains how the Mexican case study from the 1940s to the 1960s can help us understand contemporary deportation processes. Furthermore, I explain how Mexico has managed contemporary flows of deportation.

---

## Editorial Decision

unknown